# Combining Phenylalanine and Leucine Levels Predicts 30-Day Mortality in Critically Ill Patients Better than Traditional Risk Factors with Multicenter Validation

**DOI:** 10.3390/nu15030649

**Published:** 2023-01-27

**Authors:** Yi-Liang Tsou, Chao-Hung Wang, Wei-Siang Chen, Huang-Ping Wu, Min-Hui Liu, Hsuan-Ching Lin, Jung-Jung Chang, Meng-Shu Tsai, Tien-Yu Chen, Cheng-I Cheng, Jih-Kai Yeh, I-Chang Hsieh

**Affiliations:** 1Heart Failure Research Center, Division of Cardiology, Department of Internal Medicine, Chang Gung Memorial Hospital, Keelung 20401, Taiwan; 2School of Medicine, Chang Gung University, Taoyuan 33302, Taiwan; 3Intensive Care Unit, Division of Cardiology, Department of Internal Medicine, Chang Gung Memorial Hospital, Keelung 20401, Taiwan; 4Division of Pulmonary, Critical Care and Sleep Medicine, Chang Gung Memorial Hospital, Keelung 20401, Taiwan; 5Department of Nursing, Keelung Chang Gung Memorial Hospital, Keelung 20401, Taiwan; 6Division of Cardiovascular Disease, Department of Internal Medicine, Chang Gung Memorial Hospital, Chiayi 61363, Taiwan; 7Division of Cardiology, Department of Internal Medicine, Kaohsiung Chang Gung Memorial Hospital, Kaohsiung 83301, Taiwan; 8Division of Cardiology, Department of Internal Medicine, Chang Gung Memorial Hospital, Linkou 33305, Taiwan

**Keywords:** phenylalanine, leucine, prognosis, intensive care, biomarkers

## Abstract

In critically ill patients, risk scores are used; however, they do not provide information for nutritional intervention. This study combined the levels of phenylalanine and leucine amino acids (PLA) to improve 30-day mortality prediction in intensive care unit (ICU) patients and to see whether PLA could help interpret the nutritional phases of critical illness. We recruited 676 patients with APACHE II scores ≥ 15 or intubated due to respiratory failure in ICUs, including 537 and 139 patients in the initiation and validation (multicenter) cohorts, respectively. In the initiation cohort, phenylalanine ≥ 88.5 μM (indicating metabolic disturbance) and leucine < 68.9 μM (indicating malnutrition) were associated with higher mortality rate. Based on different levels of phenylalanine and leucine, we developed PLA scores. In different models of multivariable analyses, PLA scores predicted 30-day mortality independent of traditional risk scores (*p* < 0.001). PLA scores were then classified into low, intermediate, high, and very-high risk categories with observed mortality rates of 9.0%, 23.8%, 45.6%, and 81.8%, respectively. These findings were validated in the multicenter cohort. PLA scores predicted 30-day mortality better than APACHE II and NUTRIC scores and provide a basis for future studies to determine whether PLA-guided nutritional intervention improves the outcomes of patients in ICUs.

## 1. Introduction

Recent decades have seen substantial improvements in critical care. This improvement is associated with advanced therapeutic modalities and development of precise risk stratification models, such as the Acute Physiology and Chronic Health Evaluation II (APACHE II), Sequential Organ Failure Assessment (SOFA), and Nutrition Risk in the Critically Ill (NUTRIC) [1,2,3]. Nevertheless, the mortality rate among critically ill patients admitted to intensive care units (ICUs) remains high, approaching 27% in one month [4,5]. Precision nutrition therapy is essential to improve short- and long-term outcomes in different phases of the patient’s journey during critical illness [6,7]. The phases of critical illness, however, are not well-defined. A risk stratification tool for critically ill patients that can also provide precise information for nutritional intervention would be useful.

Previously, we and others found that phenylalanine can provide prognostic value in patients with infection [8], coronavirus disease 2019 (COVID-19) [9,10], cardiovascular diseases [11,12,13], or critical illness [13,14]. In adults, the elevation of phenylalanine is related to tissue breakdown [11,15], liver and kidney dysfunction [11,14,16], systemic inflammation and tremendous oxidative stress [15], as well as insufficient tetrahydrobiopterin (BH4) [14], the co-factor of the enzyme that metabolizes phenylalanine [11,14,16]. Stress hyperphenylalaninemia has been associated with a three-fold increase of mortality in ICU patients [13,14]. On the other hand, a U-shaped relationship between leucine levels and mortality in ICU patients was also noted [17,18]. High leucine levels may represent large endogenous release of energy and amino acids in the acute phase of critical illness and anabolic resistance, while low leucine levels may imply malnutrition. Although the interpretation of the changes in these amino acids needs to be further investigated, measuring them may provide information that adds to traditional mortality risk assessment and could be useful for anti-oxidant therapy and precise nutritional supplementation at the right time while avoiding over-feeding.

Thus, the aims of this study were as follows: (1) to verify the prognostic value of phenylalanine and leucine, and to investigate whether they can be combined to develop a phenylalanine leucine amino acid (PLA) score independent of traditional risk scores for predicting 30-day mortality in ICU patients; (2) to investigate whether information provided by these two amino acids and PLA scores was superior for predicting mortality to traditional nutritional biomarkers, such as albumin, pre-albumin, and transferrin; and (3) to validate use of this score in an independent cohort. The findings of this study provide a foundation for future studies to assess whether PLA-based nutritional interventions improve the outcomes of patients with critical illness.

## 2. Materials and Methods

### 2.1. Patient Enrollment

For the initiation cohort from April 2017 to November 2021, patients in critical condition were consecutively enrolled at the medical and cardiac ICUs based on these inclusion criteria: (1) they had an APACHE II score of ≥15 or were intubated due to respiratory failure; (2) they were over 20 years old; and (3) they needed to stay in the ICU > 48 h. The exclusion criteria were (1) patients who died before baseline phenylalanine measurement or (2) patients with other comorbidities that were not related to the main cause for admission and might compromise their survival in 3 months, such as cancer at terminal stage. The study was designed and carried out in accordance with the principles of the Declaration of Helsinki and had approval from the Ethics Review Board of Chang Gung Memorial Hospital. All patients provided informed consent.

For validation, an independent cohort was recruited from multiple centers, including Lin-ko, Chia-Yi, Kaohsiung, and Keelung Chang Gung Memorial Hospitals from November 2021 to August 2022 (Figure 1). The patient number ratio of the initiation cohort to validation cohort was 4:1.

### 2.2. Scoring Systems

On the day of admission to the ICU, APACHE II [1], SOFA [2], and NUTRIC [3] scores were calculated to assess disease severity.

### 2.3. Blood Sampling and Examination

In the early morning of the day after obtaining informed consent, blood samples were collected in ethylenediaminetetraacetic acid (EDTA)-containing tubes. Plasma levels of phenylalanine and leucine were measured by ultra-performance liquid chromatography (UPLC). Measurement of pre-albumin, transferrin, albumin, estimated glomerular filtration rate (eGFR), C-reactive protein (CRP), and hemoglobin was conducted in the central laboratory of each hospital.

### 2.4. Phenylalanine and Leucine Measurement

Protein precipitation in the plasma samples (100 μL) was conducted with sulfosalicylic acid (10%). After centrifugation, derivatization was performed by aminoquinolyl-carbamyl (AQC) in acetonitrile. Phenylalanine and leucine were analyzed using the ACQUITY UPLC System (Waters Corp., Milford, MA, USA), consisting of a Sample Manager, a Binary Solvent Manager, and a Tunable UV detector. To control the system and collect data, EmpowerTM 2 software was used. A 2.1 mm × 100 mm ACQUITY BEH C18 column was used for separation at a flow rate of 0.70 mL/min. The detection limit was 3.3 μM for phenylalanine and 0.9 μM for leucine. The linear range was 25–500 μM. The average intra-assay coefficient of variation was 2.6% for phenylalanine and 4.5% for leucine. The total coefficient of variation was 2.7% for phenylalanine and 4.1% for leucine.

### 2.5. Follow-Up Program

The primary endpoint was 30-day all-cause mortality. Follow-up data were obtained from hospital records, personal communication with the patients’ physicians, and telephone interviews with patients/family. Patients were prospectively followed until death or for a maximum of 30 days.

### 2.6. Development of the Phenylalanine-Leucine Amino Acid (PLA) Score

From the initiation cohort, we used the concentration of plasma phenylalanine and leucine to generate PLA scores for a 30-day mortality prediction model (Figure 1). We used Youden’s index to define the cutoff values for phenylalanine and leucine by their ability to distinguish 30-day mortality and survival. To develop the PLA score, we used Cox multivariable analysis to select the range of phenylalanine and leucine concentrations that had statistical significance for predicting 30-day mortality. Different categories of each amino acid were defined based on normal ranges and risk ranges published in our previous studies, Youden’s index, and appropriate gradients. The regression coefficient for each category was converted into points by dividing it by the smallest regression coefficient in the model [19,20]. The PLA score was the sum of the points for each category.

### 2.7. Statistical Analysis

Results are expressed as the mean ± standard deviation (SD), the median [interquartile range (IQR)], and the number (percentage) when appropriate. Data were analyzed using *t*-test or the Mann–Whitney U test. The linear trend of the data distribution across study groups was tested using linear contrast in general linear model for continuous variables and Cochran–Armitage chi-square analysis for categorical variables. Receiver operating characteristic (ROC) curves and Youden’s index were used to identify the cutoff value of variables. A univariate Cox proportional hazards model was used to determine the predictive value of variables on mortality. Kaplan–Meier analyses with a log-rank test were used to compare time-dependent outcomes. Variables with a *p* value of <0.05 in the univariate analysis were recruited in the multivariable analysis. Based on Cox multivariable analysis, we identified independent predictors of mortality by adjusting for significant variables in the univariate analysis. We calculated hazard ratios (HRs) and 95% confidence intervals (CIs). All statistical analyses were two-sided using SPSS software (version 22.0, SPSS, Chicago, IL, USA) and R software (version 3.5.1, The R foundation for Statistical Computing, Vienna, Austria). A *p* value of <0.05 was considered significant.

## 3. Results

### 3.1. Baseline Characteristics and Laboratory Data

The baseline characteristics for the initiation cohort (*n* = 537) are shown in Table 1. The mean age was 71.5 years. Of the 537 patients, 323 (60.1%) were male, 415 (77.3%) were on ventilators, and 209 (38.9%) were receiving inotropic agents. The average APACHE II, SOFA, and NUTRIC scores were 18.8, 7.1, and 5.3, respectively.

These patients were admitted to the ICU for the following conditions: 292 patients (54.4%) for cardiovascular disease [including 165 patients (56.5%) due to heart failure]; 177 patients (33.0%) for infection; 29 patients (5.4%) for pulmonary diseases; 33 patients (6.1%) for gastrointestinal bleeding; and 6 patients (1.1%) for other conditions. Phenylalanine and leucine concentrations ranged from 38.1 μM to 645 μM and from 28.3 μM to 683 μM, respectively.

During the 30-day follow-up period, 150 patients (27.9%) died. Patients in the death group showed higher likelihood of being male, having atrial fibrillation, using a ventilator, and receiving inotropic agents. Higher APACHE II, SOFA, and NUTRIC scores were also noted in the death group. Compared with the survival group, CRP levels were higher in the patients who died. The levels of cholesterol, albumin, pre-albumin, and transferrin were lower in the non-survivors (Table 1).

For the validation cohort, 139 patients were recruited from 4 centers. The validation cohort had no significant differences in the demographic and laboratory variables when compared to the initiation cohort (Appendix A). In the 30 days after enrollment, 41 patients (29.5%) died.

### 3.2. The Relationship between Mortality and Phenylalanine and Leucine Levels

In the univariate analysis for the initiation cohort, higher phenylalanine levels were linearly correlated with higher 30-day mortality (each increase of 10 μM of phenylalanine, HR = 1.09, 95% CI = 1.07–1.10, *p* < 0.001). Based on Youden’s index, the cutoff for phenylalanine was set at 88.5 μM. Phenylalanine ≥ 88.5 μM was associated with a higher 30-day mortality rate, compared to phenylalanine < 88.5 μM (HR = 3.49, 95% CI = 2.47–4.92, *p* < 0.001) (Appendix A). Next, we analyzed the prognostic value of leucine. Since a U-shaped relationship was noted in our previous study, the cutoff points for the U-curve were set at 68.9 μM and 165.6 μM, based on Youden’s index. In the univariate analysis, patients with leucine > 165.6 μM and < 68.9 μM had a higher 30-day mortality rate (HR = 1.82, 95% CI = 1.24–2.67, *p* = 0.002 and HR = 1.85, 95% CI = 1.14–3.01, *p* = 0.012, respectively), compared to patients with leucine levels between 68.9 μM and 165.6 μM. In multivariable analysis, only phenylalanine ≥ 88.5 μM and leucine < 68.9 μM were independently associated with a higher mortality rate (HR = 3.85, 95% CI = 2.67–5.55, *p* < 0.001 and HR = 2.76, 95% CI = 1.68–4.54, *p* < 0.001, respectively). In addition, we noted that, in patients with phenylalanine < 88.5 μM, a higher leucine level was linearly correlated with a lower 30-day mortality (each increase of 10 μM of leucine, HR = 0.92, 95% CI = 0.84–0.99, *p* = 0.03). Based on these findings, we built up the PLA score model.

### 3.3. The Building of PLA Score

Based on the above findings, we stratified the concentrations of phenylalanine into eight levels and leucine into four levels (Table 2). Next, we conducted Cox multivariable analysis to calculate the points for each level of amino acids. The regression coefficient (B) was determined for every level of amino acids. The smallest B (0.485) was used as base constant so that the points were derived from the formula B/0.485. Table 2 shows the points for each level of the two amino acids. The PLA score was the sum of the points of phenylalanine and leucine. In the initiation cohort, the mean PLA score was 2.75 ± 1.91 (ranging from 0 to 10.3 points).

For clinical application, patients were classified into four risk categories according to their PLA scores and the observed mortality. In the initiation cohort, 0–1 point indicated low risk; 1.1–4 points, intermediate risk; 4.1–5 points, high risk; and >5 points, very-high risk. The observed 30-day mortality rate in each risk group was 9.0%, 23.8%, 45.6%, and 81.8%, respectively. Table 3 presents the characteristics of the four risk categories. The APACHE II, SOFA, and NUTRIC scores increased as PLA scores increased. As PLA score risk categories increased, the levels of CRP became elevated, and cholesterol, albumin, pre-albumin, and transferrin declined. Figure 2A shows the Kaplan–Meier curves of the four risk groups according to PLA scores (log-rank = 167.8, *p* < 0.001). Using the low risk group as the reference for 30-day mortality, the very-high risk group had a relative risk of 19.5 (95% CI = 9.91–38.5, *p* < 0.001) (Appendix A). Based on the grid of phenylalanine and leucine concentrations, Figure 2B shows the relative risk of 30-day mortality associated with different scores compared to the reference. In the validation cohort, the Kaplan–Meier curves show that PLA score-based risk classification still significantly predicted the risk of mortality (log-rank = 19.2, *p* < 0.001) (Figure 2C).

### 3.4. The Prognostic Value of PLA Scores, Other Scores, and Biomarkers

In the univariate analysis, 30-day mortality was associated with higher PLA, APACHE II, SOFA, and NUTRIC scores, higher CRP levels, but lower levels of cholesterol, albumin, pre-albumin, and transferrin (Table 4). Since the component of NUTRIC score contains APACHE II and SOFA scores, significant collinearity exists. No collinearity was present for all other variables (all variance inflation factors were lower than 1.6). We performed two models of multivariable analysis. In model 1, when all these factors were recruited in the model (except NUTRIC score), there were only two independent predictors, including PLA (HR = 1.46, 95% CI = 1.33–1.60) and APACHE II scores (HR = 1.07, 95% CI = 1.03–1.11). In model 2 (without APACHE II and SOFA scores), there were also two independent predictors, including PLA (HR = 1.48, 95% CI = 1.35–1.62) and NUTRIC scores (HR = 1.18, 95% CI = 1.07–1.30). Based on ROC curves (Figure 2D), the area under the curves were 0.763 (95% CI = 0.725–0.799), 0.682 (95% CI = 0.640–0.721), and 0.668 (95% CI = 0.626–0.707) for PLA, APACHE II, and NUTRIC scores, respectively (PLA vs. APACHE II, *p* = 0.019; PLA vs. NUTRIC, *p* = 0.003).

## 4. Discussion

In this study, we focused on patients who were in critical condition with an APACHE II score ≥ 15 or intubated due to respiratory failure. Our study verified the predictive value of phenylalanine and leucine levels for 30-day mortality in patients in the ICU. Furthermore, we integrated these two amino acids to develop the PLA score. PLA scores predicted 30-day mortality independently of traditional risk factors, risk scores such as APACHE II, SOFA, and NUTRIC scores, and nutritional biomarkers. In addition, the prognostic value of the PLA score was superior to APACHE II, SOFA, and NUTRIC scores, as well as traditional nutritional biomarkers, including albumin, pre-albumin, and transferrin. The value of the PLA score was validated in an independent cohort recruited from multiple centers.

### 4.1. The Meaning of Elevated Phenylalanine Levels

Recent studies demonstrated that stress hyperphenylalaninemia predicts mortality in critically ill patients [9,10,13,14]. Atila et al. reported that in patients with severe COVID-19, the level of phenylalanine increased significantly compared with the control group; however, the levels of all other amino acids decreased [9,10,13,14]. Phenylalanine elevation in critical stress has been associated with depletion of BH4, a co-factor of phenylalanine hydroxylase, and with genetic variants on the genes for BH4 synthesis and recycling [14].

Sepsis or systemic inflammatory responses to infectious or non-infectious stimuli are increasingly common causes of mortality among patients in ICUs. Previously, Wischmeyer et al. [6] and Berger et al. [21] divided these patients into three phases, including acute, chronic, and recovery phases. In the acute phase, an overwhelming inflammatory mediator response to infective or non-infective stimuli results in excessive production of oxidative stress. Reactive oxygen species consume tremendous amounts of BH4 and activate the production process of BH4 either by de novo synthesis or recycling [22,23]. Genetic variants on these pathways cause inadequate repletion of BH4, leaving phenylalanine unmetabolized. Stress hyperphenylalaninemia may be used as an indicator of substantially increased oxidative stress and may indicate the timing of anti-oxidant therapy. However, the benefit of anti-oxidant therapy during the acute phase has not been well proven [24]. Whether phenylalanine level-guided anti-oxidant or BH4 supplementation is effective in lowering the mortality risk requires further investigation [21].

Although the elevation of plasma phenylalanine concentration might be a byproduct of metabolism in response to critical stress, recent reports have suggested that elevated phenylalanine is toxic to inflammatory organs [25] and that ectopic phenylalanine catabolism in the heart is associated with cardiac dysfunction [26]. On the other hand, hyperphenylalaninemia represents the decompensation of alternative phenylalanine catabolism pathways, which produce tremendous amounts of phenylalanine-derived phenolic acids with oxidative stress such as phenylpyruvate, phenyllactate, and phenylacetate [27,28,29]. Stress hyperphenylalaninemia also indicates anabolic resistance, as demonstrated by overloaded amino acid and lipid downstream metabolites in the circulation [13,18]. The perturbed phenylalanine metabolism can interfere with the production of catecholamine and cause unstable hemodynamics. Moreover, prior studies reported that inadequate BH4 bioavailability is associated with impaired immunity [30], uncoupling of nitric oxidase, and microvascular hypoperfusion [22], all related to poor outcomes.

### 4.2. The Meaning of Leucine Levels

As mentioned by Wischmeyer et al. [6], early in the acute phase of critical illness, endogenous supplies and release of energy were the main sources of energy supply. However, in the recovery phase, exogenous energy delivery becomes the primary energy source [31]. It seems that the amount of protein and non-protein calories supplied should differ in different phases. However, different guidelines have provided different suggestions [32]. In addition, reduced calorie support during the acute phase seems not appropriate for malnourished patents since they do not have the metabolic reserve to generate enough endogenous energy [3,32]. These contradictions show that we should have a scientific way to judge what phase the patients are at, which is currently performed by counting days after admission to the ICU, and better ability to assess their nutritional status as well.

A previous study reported that low total amounts of essential amino acids were associated with poor outcomes in patients with advanced disease [33]. It has been shown that leucine levels correlate well with the total amount of essential amino acids [18]. Recently, we noted a U-shaped relationship between leucine concentrations and ICU mortality [17,18]. High leucine levels represent anabolic resistance and massive endogenous release of energy in the acute phase, characterized by increased unmetabolized acylcarnitines in the circulation due to mitochondrial dysfunction with impaired beta-oxidation [15,18]. However, low leucine levels represent malnutrition and are correlated with high mortality risk two weeks after ICU admission. This study confirmed these findings with cutoffs noted at 165.6 μM and 68.9 μM, respectively. However, when phenylalanine concentrations were considered in the multivariable analysis model, the prognostic value of high leucine levels was replaced by high phenylalanine levels, suggesting that elevated phenylalanine represents metabolic disturbance along with co-factor deficiency more specifically than leucine level does. In summary, integrating leucine and phenylalanine levels can identify patients at a high risk of mortality and also help investigate how nutritional needs differ at different phases of critical illness following ICU admission.

### 4.3. Comparisons between PLA Score and Traditional Biomarkers

Our data demonstrated that each 1-unit increase in PLA score was associated with a 46% increase in 30-day mortality risk, independent of APACHE II, SOFA, and NUTRIC scores, traditional risk factors, and nutritional biomarkers. These findings support that PLA score is not a direct indicator of the nutritional status of the critically ill patients in ICUs. Although APACHE II is a traditional risk score independently associated with 30-day mortality in our multivariable model, it does not translate into therapeutic recommendations. Regarding nutritional indexes, severity of hypo-albuminemia does not further discriminate high-risk patients within the overall hypo-albuminemic ICU population in the multivariable analysis. Although pre-albumin and transferrin have been found to be prognostic for mortality, their turnover rate, half-life, and potential interference by inflammation and anemia substantially limit their universal use [34,35,36]. In the current study, all these traditional biomarkers, except for the NUTRIC score, were not significantly associated with 30-day mortality. In addition, our previous study noted that hypo-albuminemia and hypo-prealbuminemia were also observed in patients with high phenylalanine and leucine levels, and can cause incorrect interpretation of nutrition status, leading to overfeeding [13].

### 4.4. Clinical Implications of the PLA Score

Based on the scoring, we separated patients into four categories: low, intermediate, high, and very-high risk of 30-day mortality. Mortality risk increased 19.5-fold in patients with very-high risk compared to those with low risk. Based on the grid of phenylalanine and leucine concentrations, patients with very-high risk had a 92-fold increase in mortality compared to the reference patients. These findings were not only present in the initial cohort. In the multicenter cohort recruiting patients from different hospitals located from northern to southern Taiwan, the predictive value of the PLA score was repeatedly demonstrated. In addition to higher mortality risk, a higher PLA score was associated with higher APACHE II, SOFA, and NUTRIC scores, higher CRP levels, but lower cholesterol, albumin, pre-albumin and transferrin levels. In a clinical setting, when poor prognosis was identified by available risk stratification tools and malnutrition was identified by traditional nutritional parameters, PLA scores added to the accuracy of outcome prediction. In addition, since one third of patients in our study were admitted to the ICU due to heart failure and there have been a number of recent publications demonstrating the value of amino acid profiles in heart failure, PLA score is potentially applicable in this population.

The levels of phenylalanine and leucine provide clues for management in patients with critical illness at acute, chronic, and recovery phases as defined by Wischmeyer et al. [6] and Berger et al. [21]. When a leucine level is >165.6 μM, actions to avoid overfeeding should be taken. Along with a phenylalanine level of >72.2 μM or even higher, phenylalanine-free diet, BH4, micronutrients, or anti-oxidants may be considered. When a leucine level is <96.6 μM, protein supplementation should be optimized. However, in patients with high PLA score characterized by low leucine but high phenylalanine levels, phenylalanine-free high-protein diet and anti-oxidant strategies may be an option. Short-term follow-up of PLA score could appropriately fine-tune the nutritional plan. Our findings provide future perspectives to improve the outcomes of patients in ICUs by nutritional interventions based on phenylalanine and leucine.

### 4.5. Limitations

There are a few limitations in this study. First, the measurement of phenylalanine and leucine depended on a central laboratory to perform UPLC. Inter-laboratory variation should be tested for clinical application if the PLA score is to be used nationwide or worldwide. Second, UPLC is time-consuming and not appropriate for bulk samples. Other convenient methods for measuring amino acids, such as enzymatic assays, should be tested. Third, measurement of multiple amino acids, rather than leucine only, might provide additional nutritional information; however, that would substantially increase the cost and may not be clinically applicable. Finally, whether phenylalanine and leucine-based nutritional interventions can improve the outcomes of patients in ICUs needs further investigation.

## 5. Conclusions

Based on a simple amino acid profile, PLA scores predicted 30-day mortality independently of traditional risk factors, risk scores such as APACHE II, SOFA, and NUTRIC scores, and nutritional biomarkers. Future clinical intervention trials are warranted to test whether phenylalanine and leucine-based nutritional interventions can improve the outcomes for critically ill patients in the ICU.

## Figures and Tables

**Figure 1 nutrients-15-00649-f001:**
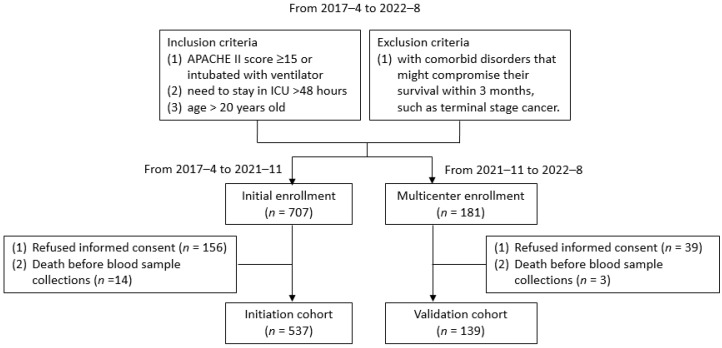
Flow diagram of the study.

**Figure 2 nutrients-15-00649-f002:**
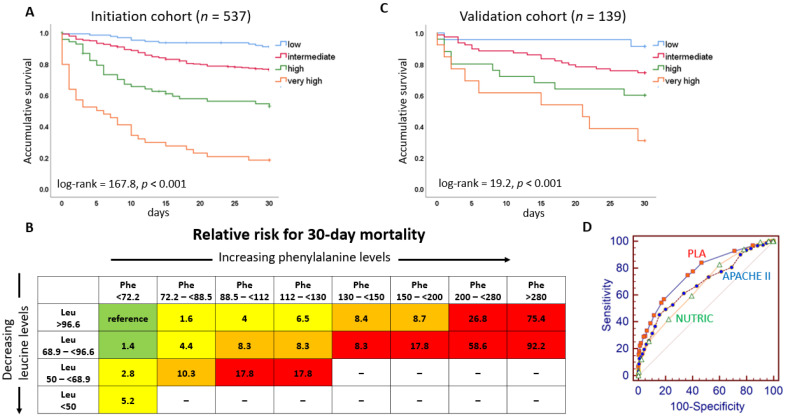
Prognostic value of phenylalanine leucine amino acid (PLA) scores. (**A**) The Kaplan–Meier curves for the four risk levels in the initiation cohort (for all-cause 30-day mortality). (**B**) Phenylalanine and leucine grid reflects the risk of 30-day mortality by intensity of coloring (green: low risk, yellow: intermediate risk, orange: high risk, red: very-high risk). Each cell represents the relative risk compared to the reference cell. “Blank” indicates the condition is not present in our study. (**C**) The Kaplan–Meier curves for the four risk levels in the validation cohort (for all-cause 30-day mortality). (**D**) The ROC curves for PLA, APACHE II, and NUTRIC scores. Abbreviations: Phe, phenylalanine; Leu, leucine; PLA, phenylalanine leucine amino acid score; APACHE II score, Acute Physiology and Chronic Health Evaluation II score; NUTRIC score, Nutrition Risk in the Critically Ill score; ROC curve, receiver operating characteristic curve.

**Table 1 nutrients-15-00649-t001:** Demographic and laboratory data for the initiation cohort.

	All	Survival	Death	
	*n* = 537	*n* = 387	*n* = 150	*p* Value
Age (years)	71.5 ± 13.6	71.4 ± 14.0	72.0 ± 12.8	0.659
Male (%)	323 (60.1)	222 (57.4)	101 (67.3)	0.034
APACHE II score	18.8 ± 5.8	17.7 ± 5.2	21.8 ± 6.3	<0.001
SOFA score	7.1 ± 3.3	6.4 ± 3.0	9.0 ± 3.5	<0.001
NUTRIC score	5.3 ± 1.9	5.0 ± 1.9	6.2 ± 1.8	<0.001
Co-morbidity				
Diabetes mellitus (%)	256 (47.7)	185 (47.8)	71 (47.3)	0.922
Hypertension (%)	350 (65.2)	249 (64.3)	101 (67.3)	0.514
Coronary artery disease (%)	231 (43.0)	172 (44.4)	59 (39.3)	0.283
Atrial fibrillation (%)	80 (14.9)	50 (12.9)	30 (20.0)	0.039
Chronic kidney disease (%) *	156 (29.1)	114 (29.5)	42 (28.0)	0.739
Ventilator use (%)	415 (77.3)	282 (72.9)	133 (88.7)	<0.001
Inotropic agent use (%)	209 (38.9)	124 (32.0)	85 (56.7)	<0.001
Days in ICU (days)	11.2 ± 7.4	11.2 ± 7.4	11.2 ± 7.2	0.957
Laboratory data				
Hemoglobin (g/dL)	10.8 ± 2.8	10.9 ± 2.8	10.6 ± 2.8	0.247
eGFR (mL/min/1.73 m^2^)	45.4 ± 43.6	47.9 ± 45.7	39.0 ± 37.1	0.020
C-reactive protein (mg/L)	46.8 (11.1–104)	36.7 (8.1–93.0)	61.7 (24.7–146)	<0.001
Cholesterol (mg/dL)	132.0 ± 52.9	138.9 ± 54.8	114.4 ± 43.2	<0.001
Triglyceride (mg/dL)	109 (81.5–152)	108 (79–150)	115 (84–158)	0.270
Albumin (g/dL)	3.19 ± 0.62	3.28 ± 0.59	2.95 ± 0.65	<0.001
Pre-Albumin (mg/dL)	15.1 ± 8.2	16.3 ± 8.7	12.0 ± 5.7	<0.001
Transferrin (mg/dL)	154.2 ± 49.9	162.3 ± 49.1	133.1 ± 45.9	<0.001

Data are expressed as the mean ± SD for variables with normal distribution, as the median [interquartile range (IQR)] for variables with skewed distribution, and as a number (percentage) for categorical variables. * Chronic kidney disease is defined as eGFR < 60 mL/min/1.73 m^2^. Abbreviations: APACHE II score, Acute Physiology and Chronic Health Evaluation II score; SOFA score, Sequential Organ Failure Assessment score; NUTRIC score, Nutrition Risk in the Critically Ill score; ICU, intensive care unit; eGFR, estimated glomerular filtration rate.

**Table 2 nutrients-15-00649-t002:** Multivariable Cox logistic regression for the PLA score calculation to predict 30-day mortality.

	B *	Points = B/0.485 **	HR (95% CI)	*p* Value
Phenylalanine (μM)				
<72.2	Reference	0	1	
72.2–<88.5	0.843	1.7	2.32 (1.26–4.27)	0.007
88.5–<112	1.556	3.2	4.74 (2.65–8.48)	<0.001
112–<130	1.793	3.7	6.01 (3.02–11.94)	<0.001
130–<150	2.096	4.3	8.13 (3.73–17.74)	<0.001
150–<200	2.403	5	11.06 (5.05–24.21)	<0.001
200–<280	3.418	7	30.50 (14.07–66.15)	<0.001
≥280	4.501	9.3	90.13 (38.62–210.37)	<0.001
Leucine (μM)				
≥96.6	Reference	0	1	
68.9–<96.6	0.485	1	1.62 (1.08–2.44)	0.020
50.0–<68.9	1.359	2.8	3.89 (2.14–7.09)	<0.001
<50	1.633	3.4	5.12 (2.12–12.36)	<0.001

* B, regression coefficient. ** Base constant B was the smallest regression coefficient in the model, which was 0.485. Abbreviations: PLA score, phenylalanine leucine amino acid score; HR, hazard ratio; CI, confidence interval.

**Table 3 nutrients-15-00649-t003:** Characteristics of patients with different PLA scores in the initiation cohort.

	PLA Score				*p* for Trend
	0–1	1.1–4	4.1–5	>5	
Variable	*n* = 122	*n* = 303	*n* = 68	*n* = 44	
Age (years)	69.8 ± 14.9	72.0 ± 13.1	73.0 ± 13.4	71.0 ± 14.1	0.529
Male (%)	64 (52.5)	192 (63.4)	38 (55.9)	29 (65.9)	0.217
APACHE II score	17.5 ± 5.6	18.3 ± 5.5	21.2 ± 5.5	22.6 ± 6.7	<0.001
SOFA score	5.5 ± 2.9	7.1 ± 3.0	8.5 ± 3.7	10.0 ± 3.5	<0.001
NUTRIC score	4.7 ± 2.0	5.3 ± 1.8	5.9 ± 2.1	6.5 ± 1.7	<0.001
Co-morbidity					
Diabetes mellitus (%)	61 (50.0)	143 (47.2)	33 (48.5)	19 (43.2)	0.515
Hypertension (%)	71 (58.2)	210 (69.3)	42 (61.8)	27 (61.4)	0.755
Coronary artery disease (%)	52 (42.6)	137 (45.2)	25 (36.8)	17 (38.6)	0.420
Atrial fibrillation	15 (12.3)	39 (12.9)	14 (20.6)	12 (27.3)	0.008
Chronic kidney disease (%) *	34 (27.9)	91 (30.0)	16 (23.5)	15 (34.1)	0.807
Ventilator use (%)	80 (65.6)	240 (79.2)	57 (83.8)	38 (86.4)	0.001
Inotropic agent use (%)	31 (25.4)	112 (37.0)	36 (52.9)	30 (68.2)	<0.001
Days in ICU (day)	10.2 ± 6.7	11.8 ± 7.6	12.1 ± 7.1	8.9 ± 7.1	0.379
Laboratory data					
Hemoglobin (g/dL)	10.8 ± 2.4	10.9 ± 2.9	10.8 ± 2.8	9.7 ± 3.1	0.025
eGFR (ml/min/1.73 m^2^)	59.6 ± 54.8	43.4 ± 40.6	38.8 ± 34.5	29.8 ± 29.6	<0.001
C-reactive protein (mg/L)	34 (7–67)	47 (12–100)	73 (18–153)	51 (16–148)	0.010
Cholesterol (mg/dL)	144.5 ± 66.8	134.6 ± 47.0	121.0 ± 43.6	94.7 ± 40.8	<0.001
Triglyceride (mg/dL)	112 (77–153)	112 (84–153)	115 (82–156)	87 (68–121)	0.104
Albumin (g/dL)	3.26 ± 0.56	3.25 ± 0.61	2.90 ± 0.63	2.97 ± 0.75	<0.001
Pre-Albumin (mg/dL)	17.1 ± 6.9	15.4 ± 6.9	11.7 ± 6.2	10.2 ± 5.8	<0.001
Transferrin (mg/dL)	160.7 ± 44.4	159.5 ± 50.0	138.1 ± 49.0	125.9 ± 48.4	<0.001

Data are expressed as the mean ± SD for variables with normal distribution, as the median [interquartile range (IQR)] for variables with skewed distribution, and as a number (percentage) for categorical variables. * Chronic kidney disease is defined as eGFR < 60 mL/min/1.73 m^2^. Abbreviations: PLA score, phenylalanine leucine amino acid score; APACHE II score, Acute Physiology and Chronic Health Evaluation II score; SOFA score, Sequential Organ Failure Assessment score; NUTRIC score, Nutrition Risk in the Critically Ill score; ICU, intensive care unit; eGFR, estimated glomerular filtration rate.

**Table 4 nutrients-15-00649-t004:** Cox univariate and multivariable analysis for predicting 30-day mortality in the initiation cohort.

	Univariate		Multivariable (Model 1) *		Multivariable (Model 2) ^†^	
	HR (95% CI)	*p* Value	HR (95% CI)	*p* Value	HR (95% CI)	*p* Value
PLA score	1.62 (1.51–1.75)	<0.001	1.46 (1.33–1.60)	<0.001	1.48 (1.35–1.62)	<0.001
APACHE II score	1.12 (1.09–1.15)	<0.001	1.07 (1.03–1.11)	<0.001		
SOFA score	1.20 (1.15–1.26)	<0.001	1.04 (0.98–1.11)	0.061		
NUTRIC score	1.34 (1.23–1.46)	<0.001			1.18 (1.07–1.30)	0.001
Age (years)	1.00 (0.99–1.01)	0.728				
Sex (male)	1.43 (1.02–2.01)	0.040	1.37 (0.96–1.96)	0.086	1.33 (0.93–1.90)	0.118
Atrial fibrillation	1.50 (1.01–2.24)	0.046	1.06 (0.69–1.62)	0.793	0.95 (0.61–1.47)	0.816
C-reactive protein (log)	1.78 (1.36–2.32)	<0.001	1.29 (0.96–1.72)	0.087	1.29 (0.96–1.74)	0.086
eGFR (mL/min/1.73 m^2^)	0.99 (0.99–1.00)	0.034	1.00 (0.99–1.01)	0.718	0.99 (0.99–1.00)	0.804
Cholesterol (mg/dL)	0.99 (0.98–0.99)	<0.001	0.99 (0.99–1.01)	0.551	0.99 (0.99–1.00)	0.261
Albumin (g/dL)	0.48 (0.37–0.62)	<0.001	0.88 (0.64–1.20)	0.411	0.85 (0.62–1.15)	0.284
Pre-Albumin (mg/dL)	0.92 (0.89–0.94)	<0.001	0.99 (0.96–1.02)	0.626	0.99 (0.98–1.02)	0.937
Transferrin (mg/dL)	0.99 (0.98–0.99)	<0.001	0.99 (0.99–1.01)	0.277	0.99 (0.99–1.00)	0.117

* Model 1 includes all significant variables in the univariate analysis, except NUTRIC score; ^†^ Model 2 includes all significant variables in the univariate analysis, except APACHE II and SOFA scores. Abbreviations: HR, hazard ratio; CI, confidence interval; PLA score, phenylalanine leucine amino acid score; APACHE II score, Acute Physiology and Chronic Health Evaluation II score; SOFA score, Sequential Organ Failure Assessment score; NUTRIC score, Nutrition Risk in the Critically Ill score; eGFR, estimated glomerular filtration rate.

## Data Availability

The datasets used and analyzed during the current study are available from the corresponding author upon reasonable request.

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
