# Peer review of "Combining Phenylalanine and Leucine Levels Predicts 30-Day Mortality in Critically Ill Patients Better than Traditional Risk Factors with Multicenter Validation"

_nutrients, 2023, doi:10.3390/nu15030649_

Round 1

Reviewer 1 Report

The authors examined the predictive ability of combined levels of phenylalanine and leucine amino acids (PLA) in the initiation and validation cohorts of critically ill patients in the intensive care unit (ICU) for 30-day mortality.

The results showed that the prognostic value of the PLA score predicted 30-day mortality better than conventional nutritional biomarkers such as APACHE II score, SOFA score, albumin, prealbumin, and transferrin.

I believe that the results of this study are of great interest and academic significance.

1) Patients admitted to the ICU have a variety of medical conditions, of which heart failure is one of the major important ones. In the text, you mention of the 537 patients, 292 patients (54.4%) for cardiac reasons.

What percentage of these patients had heart failure?

The authors focus on phenylalanine and leucine in particular in the amino acid profile, and there have been a number of recent papers focus on heart failure and amino acid profiles (ESC Heart Fail. 2022 Oct 27.   Heart Vessels. 2021 Jul;36(7):965-977.   J Cardiol. 2020 Jun;75(6):689-696.).

Please add a discussion of the applicability of this PLA score to heart failure patients.

2) In this study, all conventional nutrition-related biomarkers were not significantly associated with 30-day mortality. This suggests that the short-term prognosis of ICU patients is not defined by their nutritional status (malnutrition), and the reader may better understand the comparison between the PLA score and conventional nutritional indices if it is emphasized that the PLA score is not a direct indicator of the nutritional status of the patient.

3) Based on the results of this study, it would be helpful to discuss in more detail, if possible, future perspectives on whether nutritional interventions based on phenylalanine and leucine can improve the outcomes of ICU patients.

Reviewer 2 Report

The present study by Tsou et al investigated the prognostic ability of the levels of phenylalanine and leucine amino acids (PLA) or the composite PLA score in ICU patients. Overall the study is interesting and well written. However, I have one major comment.

The authors claim that they wanted to “investigate whether information provided by these two amino acids and PLA scores was superior for predicting mortality to traditional nutritional biomarkers, such as albumin, pre-albumin, and transferrin”. First of all, in the ICU population albumin, pre-albumin, and transferrin levels serve not only as nutritional indexes but primarily as acute phase factors that correlate with the degree of the inflammatory process. Surprisingly, the authors have not used any nutritional tools. It is true that most of the traditional screens and assessments are often limited due to their subjective nature. However, NRS 2002 and the Nutrition Risk in the Critically ill (NUTRIC) have been extensively studied. In fact, the Nutrition Risk in Critically Ill (NUTRIC) Scoring System has emerged as a useful tool for nutrition risk assessment but also, and more importantly, it is independently related to the risk of 28-day mortality in ICU patients. The NUTRIC Score has been shown to identify patients who are particularly likely to have a mortality benefit from aggressive nutrition therapy. Why did n’t the authors use such a composite score to describe the nutritional status of their cohort and to compare its prognostic ability with the PLA score.  

Round 2

Reviewer 2 Report

All my comments have been addressed adequately. I have no further comments.